# Multidisciplinary Approach to Spinal Metastases and Metastatic Spinal Cord Compression—A New Integrative Flowchart for Patient Management

**DOI:** 10.3390/cancers15061796

**Published:** 2023-03-16

**Authors:** Miguel Esperança-Martins, Diogo Roque, Tiago Barroso, André Abrunhosa-Branquinho, Diogo Belo, Nuno Simas, Luis Costa

**Affiliations:** 1Medical Oncology Department, Centro Hospitalar Universitário Lisboa Norte, 1649-028 Lisboa, Portugal; 2Luís Costa Lab, Instituto de Medicina Molecular—João Lobo Antunes, Faculdade de Medicina da Universidade de Lisboa, 1649-028 Lisboa, Portugal; 3Sérgio Dias Lab, Instituto de Medicina Molecular—João Lobo Antunes, Faculdade de Medicina da Universidade de Lisboa, 1649-028 Lisboa, Portugal; 4Neurosurgery Department, Centro Hospitalar Universitário Lisboa Norte, 1649-028 Lisboa, Portugal; 5Radiotherapy Department, Centro Hospitalar Universitário Lisboa Norte, 1649-028 Lisboa, Portugal

**Keywords:** spinal metastases, metastatic spinal cord compression, spinal stability, radiosensitivity, prognosis, multidisciplinary

## Abstract

**Simple Summary:**

The prevalence of metastatic spine disease is progressively increasing, affecting a growing group of heterogeneous and complex patients. A multidisciplinary, personalized approach, enriched by the expertise of each involved specialty (namely oncologists, radiotherapists, neurosurgeons, orthopedic surgeons, hematologists, and neuroradiologists), is pivotal and achieves superior results in terms of clinical outcomes. We reviewed the most recent data regarding the pathophysiology of metastatic spine disease, prognostic scores, and available treatment options and formulated a proposal for an updated algorithmic approach to the pathology according to the clinical scenario of each patient. A flowchart-based approach to patients offers an evidence-based management of metastatic spine disease, providing a valuable clinical decision tool in a context of high uncertainty and quick-acting need. Nevertheless, we underline that the goal of this type of approach is to assist in clinical decisions, not to replace a case-by-case reflection concerning the specificities of each patient.

**Abstract:**

Metastatic spine disease (MSD) and metastatic spinal cord compression (MSCC) are major causes of permanent neurological damage and long-term disability for cancer patients. The development of MSD is pathophysiologically framed by a cooperative interaction between general mechanisms of bone growth and specific mechanisms of spinal metastases (SM) expansion. SM most commonly affects the thoracic spine, even though multiple segments may be affected concomitantly. The great majority of SM are extradural, while intradural-extramedullary and intramedullary metastases are less frequently seen. The management of patients with SM is particularly complex and challenging, with multiple factors—such as the spinal stability status, primary tumor radio and chemosensitivity, cancer biological burden, patient performance status and comorbidities, and patient’s oncological prognosis—influencing the clinical decision-making process. Different frameworks were developed in order to systematize and support this process. A multidisciplinary, personalized approach, enriched by the expertise of each involved specialty, is crucial. We reviewed the most recent evidence and proposed an updated algorithmic approach to patients with MSD according to the clinical scenario of each patient. A flowchart-based approach offers an evidence-based management of MSD, providing a valuable clinical decision tool in a context of high uncertainty and quick-acting need.

## 1. Background

Spinal cord compression is counted among the classical cancer-related emergencies [1,2]. When not treated quickly, it can lead to permanent neurological damage and long-term disability. Even without spinal cord compression, spinal metastases can cause symptoms that lead to significant disability and loss of quality of life [3,4].

A recent systematic review estimated that the overall cumulative incidence of spinal metastases and metastatic epidural spinal cord compression is, respectively, 15.67% and 2.84% in patients with a solid malignancy [5].

Detailed analyses and estimations of the direct impact of metastatic spinal cord compression on the overall survival of cancer patients are scarce.

Telles da Silva et al. performed a cohort study including 1112 patients with non-small-cell lung cancer and reported that, among patients with non-small-cell lung cancer, the ones that presented with metastatic spinal cord compression were 1.43 times more likely to die than those with no history of metastatic spinal cord compression [6]. Besides this, the median survival time was 8.04 months for those who presented metastatic spinal cord compression and 11.95 months for those who did not present metastatic spinal cord compression [6].

He et al. studied patients with metastatic spinal cord compression secondary to primary hepatocellular carcinoma and described a median progression-free survival of 7.0 months and a median overall survival of 9.7 months within this population [7].

Other studies reported the deleterious effects of metastatic spinal cord compression on the survival of patients with prostate cancer, cancer of unknown primary, and multiple myeloma [8,9,10].

Rades et al. developed the first overall survival predictive score for patients with metastatic spinal cord compression [11], while Nenclares et al. recently created and presented a scoring system to predict overall survival in patients that have been irradiated for metastatic spinal cord compression [12].

The tumors with the highest prevalence of bone involvement are the solid tumors arising from the breast (70% of all metastatic breast cancer), prostate (85%), lung (40%), and kidney (40%), as well as multiple myeloma (95%) [13]. Vertebral metastases are very frequent among patients with cancer, and metastatic spinal cord compression will occur in up to 20% of patients with vertebral metastases [1]. A series from the end of the decade 1990–2000 reported that pain from fracture or metastatic spinal cord compression can be the first symptom of cancer in 20% of cases [2]. The number of studies that specifically evaluated the prevalence of metastatic spinal cord compression as the first manifestation of cancer is scarce. A recent study estimated that metastases originating from cancers of unknown primary account for around 10% of the totality of cases of metastatic spinal cord compression [9].

### 1.1. General Pathophysiological Mechanisms of Bone Metastases

Metastatic bone disease is not a simple mechanical process resulting from the embolization of tumor cells. The formation of bone metastases depends on reciprocal interactions between the cancer cells and the stromal and immune cells in the bone microenvironment. This unique microenvironment is highly susceptible to tumor seeding. The key players in the cancer-bone interactions are the osteocytes (embedded within the bone matrix), the local immune cells, the osteoblasts (responsible for bone matrix deposition), and the osteoclasts (responsible for bone matrix reabsorption). Circulating pro-inflammatory cytokines such as TGF-beta, interleukin-6, and interleukin-1, as well as endocrine hormones such as estradiol and paracrine signaling molecules such as osteoprotegerin, mediate interactions between osteoblasts and osteoclasts, thus regulating bone homeostasis. Some of these interactions are mediated by the RANK/RANK-ligand system, which can be inhibited in vivo with the monoclonal antibody denosumab, with clinically proven results in metastatic bone disease [13,14].

Different cancer types have different impacts on this homeostatic balance. Tumors can create *blastic metastases* (e.g., breast and prostate cancer), in which bone matrix deposition is dominant. These kinds of metastases are radio-opaque due to the increase of bone matrix deposition. Fractures are less frequent with these metastases, but bone pain is frequent. On the opposite end of the spectrum, tumors can create *lytic metastases* (e.g., breast, lung, and renal cancers as well as myeloma bone disease), in which bone resorption is dominant. With few exceptions (such as kidney cancer and multiple myeloma), and despite the dominance of bone matrix resorption, lytic bone lesions display substantial osteoblast activity, which can be shown in vivo through nuclear radioisotope imaging. These metastases are radiolucent due to a lower density of bone matrix and have a higher risk of fracture with minimal trauma. Most solid tumor metastases lie somewhere on the blastic-lytic spectrum, have simultaneous bone destruction and bone formation, and are aptly named *mixed metastases*. On the other hand, as mentioned above, kidney cancer metastases and myeloma bone disease usually give rise to exclusively lytic bone lesions. This is due to the suppression of osteoblast activity by the tumor. In the case of multiple myeloma, myeloma cells secrete soluble mediators such as Dickkopf-related protein 1 (DKK1), activin A, and soluble frizzled-related protein 3 (sFRP3), which directly suppress osteoblastic bone formation [13].

The central role of the osteoclast in bone metastases and myeloma bone disease has led to the use of bone-targeted agents in the setting of metastatic bone disease. Two main pharmacological classes are available, with clinically proven efficacy: the inhibitors RANK/RANKL (i.e., the monoclonal antibody denosumab), which inhibit osteoclast function through disruption of this signaling pathway, and the bisphosphonates (e.g., alendronic acid and ibandronic acid, among others), which are taken up by osteoclasts and directly inhibit bone resorption. Bisphosphonates can be further divided into nitrogen-containing bisphospohnates (e.g., zoledronate, ibandronate, and pamidronate) and non-nitrogen-containing bisphosphonates (e.g., clodronate), which act through different intracellular pathways. The common side-effects of these drugs result from inhibition of bone resorption: hypocalcemia and, more rarely, insufficiency fractures. Dosing must be adjusted for renal function. [13]. Despite the continued evolution of systemic therapy in the last three decades, studies continue to show benefits in the use of bone target agents [13], and their use is advocated in recent guidelines [15]. Both denosumab and bisphosphonates shown efficacy in reducing skeletal adverse events, reducing pain, and improving quality of life. No drug has shown clear superiority against the others (the reader is directed to [13] for a more extensive treatment of the cellular and molecular pathways involved and references to specific clinical trials).

### 1.2. Specific Pathophysiological Framework of Spinal Metastases and Metastatic Spinal Cord Compression

When taken as a whole, the spine is the most frequent site of bone involvement, with around 87% of bone metastases localizing to the spine. Bone metastases are more common in the axial skeleton (spine, pelvis, and skull), as well as in the trabecular bone of the proximal humerus and femur. They are rare in the predominantly cortical bones of the appendicular skeleton (1% of bone metastases). The rich blood supply to the spine and the high amount of medullary bone, which acts as the main metastatic niche for tumor cells, probably explain the high prevalence of spinal metastases [13,14]. At the moment, it is not clear whether the molecular microenvironment of the vertebrae creates a specific tropism different from the medullary bone at other locations, or whether the higher prevalence of vertebral metastases is only due to the vascular factors and large amount of medullary bone, as described above. In any case, it is clear that the probability of vertebral metastases is more dependent on the origin of the primary tumor than on the proximity of the tumor to vascular structures such as venous plexuses and terminal arteries (e.g., vertebral metastases from breast cancer are more frequent than vertebral metastases from rectal cancer, despite the close relation between the venous drainage of the rectum and the venous plexuses of the sacral and lumbar spine).

Spinal cord compression most frequently develops in a gradual and progressive way, as a consequence of the compressive effect of an enlarging vertebral mass on the spinal vasculature, thecal sac, and spinal cord [16]. Destruction and fragmentation of the cortical bone of the vertebral body may also lead to a compression deformity, either by direct spinal cord compression by vertebral collapse or by protrusion of displaced bony fragments into the epidural space [16].

The spinal cord compression process is not simply a physical phenomenon, but a temporal and pathophysiological continuum that may be conceptually divided in two phases: the primary and the secondary cord injury [17].

Primary cord injury corresponds to the mechanical trauma from compression, developing immediately after the injury, leading to physiological, cellular, and biochemical changes that result in obliteration of the neural parenchyma, interruption of the axonal network, hemorrhage, and destruction of the glial membrane [18].

Subsequently, the panoply of biological modifications within neural tissues elicited by the primary injury leads to an array of events that are related to the secondary injury and that perpetuate this loop of neuronal damage [18].

The secondary injury is a product of the continued and persistent chemical and mechanical lesion of spinal tissues. This ultimately conducts to neuronal excitotoxicity, increased reactive oxygen concentrations and glutamate levels, and subsequent disassembly of underlying nucleic acids, proteins, and phospholipids [18].

The secondary injury may thus be itself divided in three phases: acute, sub-acute, and chronic injury. The acute secondary injury phase is characterized by clinical features such as vascular damage (first manifested by spinal ischemia and then by vasogenic edema), glutamate excitotoxicity, and ionic imbalance [18]. In the case of the persistence of acute secondary injury, sub-acute secondary injury takes place and is manifested by events such as mitochondrial phosphorylation, increased calcium influx, increased calcium cytosolic levels, free radical oxygen species production, lipid peroxidation, protein damage, and the establishment of a neuroinflammatory loop [18]. The persistence of damage then ultimately leads to the chronic phase, characterized by neuronal apoptosis, acute axonal degeneration, remodeling and demyelination (with subsequent axonal dieback or Wallerian degeneration), and glial scar development [18]. Finally, the formation of a cystic cavity and the maturation of the glial scar are typically observed [18].

### 1.3. Topographical Distribution of Spinal Metastases

Spinal metastases may affect all segments of the spine. They are most common in the thoracic spine, followed by the lumbosacral spine, and the cervical spine. Bone metastases commonly affect multiple segments [19]. The topographical distribution of spinal metastases is along spinal segments, and the correspondent neurological signs and symptoms that may be associated with the neurological damage imposed by the spinal metastases are schematically illustrated in Figure 1.

### 1.4. Anatomical Classification of Spinal Metastases

Spinal metastases may spread through diverse pathways, comprising venous hematogenous spread from Batson’s venous plexus, arterial spread, direct tumor extension, lymphatic dissemination, and by subarachnoid intracanalicular and leptomeningeal seeding of primary and secondary central nervous system malignancies [21].

The seminal study of Batson, published in the 1940s [22], and later supported by the study of Coman and DeLong [23], proposed the venous plexus of Batson, an avalvular complex system of veins, located in the epidural space and particularly sensitive to the variations of blood flow and pressure in the system of the vena cava, as the main route of venous hematogenous dissemination [24]. Although Batson’s theory proposed a particularly compelling correlation between the spatial pattern of metastatic spread in the spine and the conceptually expected mechanism of hematogenous dissemination by arterial or venous routes, a correlation between tumors with proposed arterial/venous pathways of metastasis and the central/peripheral location of metastatic lesions could not be found, highlighting the existence and prominence of other mechanisms than pure arterial or venous dissemination by hydraulic gradients, such as tissue specificity, and closed loop circulation systems as stated above [25].

Nowadays, it is believed that the center of the vertebral body is the primordial niche for the development of the complex interactions between the metastatic seeds and the vertebral bone soil, with subsequent posterior spatial dissemination, characterized by the involvement of the pedicles [21]. A recent French study tried to portray the spread profile of spinal metastases and better characterize the spatial distribution of spinal metastases, describing a more frequent involvement of the vertebral body relative to the posterior elements, the presence of a circumferential spine involvement (body and posterior elements) in around one third of cases, and the existence of an associated epidural compression in half of cases [26].

Spinal metastases may be anatomically classified as extradural, intradural-extramedullary, and intramedullary metastases, according to their location and extension [27]. The vast majority (85–90%) of vertebral metastases are extradural [27].

## 2. Relevance of Spinal Metastatic Disease, Importance of Its Multidisciplinary Management, and Motivation for the Development of the Algorithms

The burden of disease attributable to metastatic bone disease is high, and most of the decrease in quality of life can be traced to skeletal-related events, namely bone pain, fractures, and metastatic spinal cord compression [28]. Among metastatic bone disease, spinal bone disease is especially relevant due to the proximity to vital neurological structures which govern sensitivity, pain perception, movement, and autonomic functions such as breathing, cardiovascular response to stress, and sphincter control.

Pain related to metastatic spine disease is the most prevalent symptom and often precedes the development of neurological signs [29]. Radicular distribution of pain or paresthesia to a specific dermatome can guide the clinician to investigate specific topographies of spine involvement [29]. Other pain-associated red flags that should warrant an urgent investigation of the potential metastatic spine involvement are: (1) progressive pain despite medical treatment; (2) worsening of pain while standing or sitting; and (3) pain preventing the patient from sleeping [29]. Motor disfunction frequently accompanies pain, and patients commonly describe rapid-onset limb weakness limiting orthostatism and ambulation in the last few days or weeks [29,30]. Autonomic dysfunction, namely urinary retention or incontinence and constipation, tends to occur in advanced cases of spinal cord compression and is uncommon without other signs and symptoms [29]. The expected clinical manifestations per spine segment involved are illustrated in Figure 1.

Spinal cord compression or spinal root compression due to fracture or direct tumor invasion can be the inaugural manifestations of metastatic disease and even the first manifestation of cancer itself. A multidisciplinary approach is the hallmark of the current approach to cancer diagnosis and treatment, and spinal metastatic disease is no exception [31,32].

A multidisciplinary team plays essential roles at three distinct moments (as summarized in Table 1):*Before spinal cord compression*, radiooncologists and medical oncologists cooperate to achieve control of the primary tumor and metastases (including the use of bone targeted agents, as mentioned in Section 1.1), in order to prevent complications. Spinal surgeons can stabilize the relevant spinal segments preemptively;During *acute spinal cord compression* or after an unstable vertebral fracture, there is a short time window in which surgical decompression and stabilization can restore a degree of functionality. In this case, an optimized referral pathway and well-established lines of communication between professionals (emergency physician, medical oncologist, radio-oncologist, and spinal surgeon) are essential;*In the post-acute phase*, after an established neurological deficit, surgical decompression, and stabilization (if not performed before) may still play a role. Radiation therapy is almost universally recommended, and systemic treatment is often indicated. Because permanent neurological sequelae are expected, other professionals, such as palliative care physicians and formal or informal caregivers, will play a role in trying to avoid further loss of quality of life. Rehabilitation plays an important role, as function recovery after metastatic spinal cord compression is similar to the one observed in traumatic spinal injury [31].

The time-sensitive nature of the neurologic deficits from spinal cord compression requires quick decisions, often under great uncertainty. As stressed above, spinal cord compression can be the first manifestation of a previously unknown cancer in up to 20% of patients [2], and even if the cancer is known, those responsible for providing urgent care in the acute setting might not have the knowledge about the primary disease process in order to accurately estimate patient prognosis and choose the optimal treatment. In cases of high uncertainty and where there is a need to act quickly, the development of decision algorithms and standardized care pathways is desirable. This need has been recognized in the form of algorithmic approaches such as the Advanced Trauma Life Support (ATLS) course for the acute management of trauma patients [33], the guidelines proposed by the European Cardiology Society for the diagnosis and treatment of acute myocardial infarction [34], and also by the American Stroke Association for the diagnosis and early management of stroke [35]. We claim that, just as in these pathological contexts, an algorithmic approach to the management of metastatic spinal disease and spinal cord compression can lead to improved clinical results. Such a systematized approach could help reduce the socioeconomic and racial disparities in order to properly care for this vulnerable population, which have been documented in the United States [36].

## 3. The Art of Prognostication, Clinical Judgment, and Decision

### 3.1. Ingredients for an Appropriate Decision: Spinal Stability Status, the Concept of Tumor Radiosensitivity, Clinical Criteria, and Expected Treatment Complications

Multiple factors must be taken into account to decide on the most appropriate management. Spinal stability status assessment and neurological deficit, tumor expected treatment response such as to radiotherapy and chemotherapy, available resources, and patient clinical features (e.g., cancer stage and biological burden, previous comorbidities, expected survival), the latter being the most weighting factor to consider, are variables of paramount importance for this decision process. Most decision tree proposals rely on the patients’ fitness degree for a surgical procedure or on the level of potential medium-to long- term benefit of any treatment [37]. These data are taking into account that all patients should be offered radiotherapy after surgical stabilization based on the Patchell et al. randomized trial [38] and that radiotherapy should be performed within 1–2 weeks (or when wound healing is complete) [39], implying the existence of a low risk of postoperative complications. It is also important to remind readers that most radiotherapy randomized trials for spinal cord compression used different fractionation regimens with overall treatment times from 1–2 days to 2–4 weeks, response assessment 4 weeks after first treatment, and that most patients had a minimum median overall survival of 4–6 months [40,41,42,43,44].

Some clinicians argue that previous data on randomized trials with surgery and/or radiotherapy and predictive factors are outdated, and new approaches should be explored with the introduction of new systemic treatments (i.e., immunotherapy in non-small-cell lung cancer and malignant melanoma) that alter the overall survival and surgical and radiotherapy techniques (e.g., stereotactic radiotherapy) that can be used in oligometastatic patients. Therefore, some authors argue that neurologic symptom degree and onset are extremely relevant in a certain niche of patients with good performance status, who are almost naive to a wide variety of systemic treatments that could prolong survival. The neurological assessment based on both clinical (e.g., American Spinal injury association international classification system) [45] and imaging evaluation (Bilsky Score) [46], along with spinal stability and tumor radiosensitivity, could dictate the best approach. It should be stressed out that no technique of radiotherapy nor fractionation regimen can treat spinal instability/fracture. In addition, there is not yet reliable comparative data to determine optimal recalcification and prevention of pathological fracture rates within radiotherapy regimens because it might be dependent on tumor histology and combined systemic treatment (e.g., bone-modifying agents) [47,48].

Expected possible treatment complications must also be considered in the decision process. Regarding spinal surgery, complications can occur, namely durotomies resulting in pseudomeningoceles and/or cerebrospinal fluid leaks, intraoperative spinal cord hypoperfusion resulting in neurological deterioration, wound infections, postoperative hematomas resulting in spinal cord and/or nerve root compression, and also instrumentation failures such as screw pullout or pseudoarthrosis development [49,50,51]. If recognized during the procedure, durotomies should always be addressed with an attempt to primary repair to reduce postoperative burden [49]. However, in cases of spinal metastatic disease, incidental dural tears that can be unrecognized during the surgery are a possibility, owing to neoplastic adhesion to the dura and the nerve roots [49]. These clinical events can undermine the postoperative period, motivate a new surgical intervention with associated comorbidity, lengthen the admission period in the healthcare unit, and delay discharge from the hospital, which can jeopardize the final outcome of the already fragile population of patients with spinal metastatic disease [49,50,51]. Neurological complications such as myelopathy from radiation are late, rare events and are normally mitigated during RT planning [52]. The estimated risk of spinal cord myelopathy a with conventional fractionated regimen (1.8–2Gy per fraction) increases if the maximum dose goes beyond 50 Gy (0.2% for 50 Gy, 6% for 60 Gy and 50% for 69 Gy). In the case of SBRT techniques (1 to 5 fractions with a high dose per fraction), Sahgal et al. proposed the following maximum dose to the spinal cord in order to reduce the risk of myelopathy from 1% to 5%: 12.4–14.0 Gy in 1 fraction; 17.0 Gy in 2 fractions, 20.3 Gy in 3 fractions, 23.0 Gy in 4 fractions, and 25.3 Gy in 5 fractions [53]. Vertebral fractures are uncommon with conventional radiotherapy techniques, as mentioned previously.

### 3.2. Evaluating the Spinal Stability Status: The Spinal Instability Neoplastic Score (SINS)

Bone destruction and volumetric expansion of the tumoral lesion may lead to the collapse and compression of the of the neurological structures in and close to the spinal and nerve root canals [54]. This deformity ultimately ends in spinal instability [54]. The reversibility of the deficits depends on the time of onset and on the degree of preservation of neurological function [54].

Spinal instability is itself a poorly defined concept, and the precise estimation of the degree of spinal instability is not easy to determine. An appropriate evaluation of the spinal instability degree is critical for the clinical decision-making process for patients with metastatic spinal lesions. Spinal instability may then be considered as the loss of spinal integrity under physiological loads, leading to pain, deformity, and/or neurological compromise under such physiological loads.

The Spinal Instability Neoplastic Score (SINS), published in 2010, is a reliable and predictive tool for clinicians, guiding their decisions regarding the benefits of a surgical intervention in patients with primary or secondary neoplastic involvement of the spine [55].

It assesses and scores six variables, namely the location of the lesion, characteristics of pain, type of bony lesion, radiographic spinal alignment, degree of vertebral body destruction, and involvement of posterolateral spinal elements [56]. The scores for each variable are added, resulting in a final score ranging from 0 to 18 [55,56]. Stability is defined by a score of 0 to 6, possibly impending instability is determined by a score of 7 to 12, while instability is coined by a score of 13 to 18 [55,56].

Patients with SINS scores greater than six should be offered a surgical consultation, but the decision-making process is particularly challenging for the majority of patients, that display scores between 7 and 12, and, subsequently, have lesions labelled as “potentially unstable” [56]. The prognostic value of SINS for patients with score values varying between this range of values (7–12) is controversial, having fueled the development of different studies that analyzed the subgroup of patients inserted in this “grey zone”. Some of these patients were actually subjected to surgical fixation after an initial conservative approach [56]. As a product of these studies, a new SINS score cutoff of 11 has been defined as indicative of possible spinal instability and of subsequent likely benefit from spinal surgical stabilization [56].

SINS also have limitations of different types, the biggest of which is the absence of modifiers for multiple spinal lesions (which is actually the most common scenario) [56].

Even though SINS has fragilities, it served as the basis for the development of the new location, mechanical instability, neurology, oncology, and patient’s features (LMNOP) approach (further explored in detail) and the Oswestry Spinal Risk Index [56].

Machine learning algorithms and artificial intelligence-based survival predictive models have already shown promising predictive quality, in comparison with traditional risk scores, and may play a key role in the future [56].

### 3.3. Tumor Radiosensitivity

Each cancer type responds differently to non-surgical treatments, such as radiotherapy and/or systemic treatments, and ongoing predictive factors are known or under research. In terms of basic radiobiology concepts, cancer cells and healthy tissues have different intrinsic fractionation radiosensitivity, which is expressed with the term alpha-beta (α/β) ratio value, which is obtained through the Linear Quadratic (LQ) Model and describes the rate of clonogenic survival of certain types of cells to an amount of radiation dose per fractionation [57]. Although the LQ model is a useful tool for predicting the equivalent total dose of different fractionation schedules and estimation of radiotherapeutic outcome, the (α/β) ratio is measured in vitro in cell line cultures or based on available clinical data from large randomized trials, but may not be representative for full clinical radiobiological effect (“radioresponsiveness”) due to other factors (e.g., cell repair, repopulation, redistribution, reoxygenation, microenvironment) [58,59,60,61,62]. In a broad sense, cells with a high α/β ratio (~10) suffer more lethal effects from small doses of radiation per fraction (e.g., myeloproliferative neoplasms, myeloma, germ cell tumors), while cells with a low α/β ratio (~1–3) are more resistant to small doses of radiation per fraction and may require moderate to extreme hypofractionation to express the lethal effect (e.g., renal cell carcinoma, malignant melanoma).

In clinical practice and relying on 3D radiotherapy techniques (3D-CRT) with the most frequent hypofractionation regimen published data (e.g., single fraction 8–10 Gy, 20 Gy in 5 fractions, 30 Gy in 10 fractions), cancers are classically categorized in “radioresponsiveness” as described in Table 2 [63], and in accordance with clinical endpoints such as pain relief, ambulatory/mobility recovery, and survival. Table 2 highlights the discrepancy between responsiveness to radiation and the intrinsic radiosensitivity (in α/β) [58,60,61].

This categorization should be revised in the future due to the emergence of new systemic treatments (e.g., targeted therapy and immunotherapy) and radiotherapy treatments (e.g., stereotactic radiotherapy techniques), in which the data is not yet robust for spinal cord compression [64] and there are conflicting results (i.e., pain relief) for the use of SBRT in uncomplicated bone metastases [65,66,67,68,69]. In addition, the risk of an induced vertebral compression fracture with spinal SBRT ranges from 11 to 39% compared with <5% with 3D-CRT [70]. In current practice, SBRT is not yet recommended due to a lack of evidence for an efficacy and safety profile and logistical/resource challenges nowadays as an emergency treatment.

The SABR-COMET phase II trial observed OS improvement with the employment of SBRT in patients in the oligometastatic setting compared to standard-of-care for each specific carcinoma. Unfortunately, this trial avoided including bone metastasis within 3 mm of the spinal cord [71]. The PRE-MODE Trial [64] was a phase II trial conducted in 40 patients who were not fit for surgery for spinal cord compression due to low performance status (*n* = 26), had poorer survival according to Rades et al. criteria (*n* = 10), were medically inoperable (*n* = 2), or had multiple myeloma (*n* = 2). Sixteen patients (40%) were not ambulatory prior to radiotherapy, and nine patients (22.5%) did not receive corticosteroids. The experimental prescription used was 25 Gy in five fractions and labelled as “precision radiation therapy” techniques (the majority being volume-modulated arc therapy, *n* = 38) to explain the maximum dose constraint applied to the spinal cord (101.5% of the prescription dose). The experimental data outcomes were compared to a historical control group of patients treated with 20 Gy in 5 fractions in conventional RT techniques, and it was assumed that the control group would have inferior local progression-free survival. Improvement was observed in terms of the ambulatory recovery rate after radiation therapy (*n* = 33, 82.5%) and in six-month survival rates (95% for LPFS and 42.6% for OS) with a low toxicity profile (*n* = 1 for grade 3, *n* = 3 for grade 2). When applying propensity score analysis for comparison with the historical control group (*n* = 664 patients), the 5 Gy in five fractions were significantly superior to the 4 Gy in five fractions with regard to LPFS (*p* = 0.026), but not motor function (*p* = 0.51) or OS (*p* = 0.82).

When patients are submitted to radiotherapy as a monotherapy treatment, the Rades criteria (Table 3) can be applied to predict for ambulatory/mobility rate after RT. The criteria were based on a retrospective a multivariate analysis of 2096 patients into 5 prognostic factors (Table 3) [72]. The initial version of the Rades criteria model was straightforward: scoring points were calculated by observing their respective ambulatory rate and divided by 10 (rounded up), which ended up with minimum score points of 21 and a maximum of 44 points. Afterwards, groups of prognostic value were set up based on the incremental total sum of the scoring points. Initially, there were five groups (with incremental changes of better ambulatory recovery): Group A (21–28 points), Group B (29–31 points), Group C (32–34 points), Group D (35–37 points) and Group E (38–44 points). This initial model was validated in two prospective cohorts [73]: 653 patients were treated only with radiotherapy, and 104 patients received surgery followed by radiotherapy. For simplification, the model was rearranged into three prognostic groups: Group I (up to 28 points), Group II (between 29–37 points), and Group III (beyond 38 points). The ambulatory outcome for the prospective RT-only cohort was lower in Group I (10.6%), and higher in Group III (98.5%). These rates were similar to the retrospective cohort: 6.2% (Group I), 68.4% (Group I), and 98.7% (Group III). When assessing patients submitted to surgery followed by radiotherapy patients (*n* = 104), two distinct cohorts were observed: the laminectomy and postoperative radiotherapy cohort vs. the laminectomy plus vertebral stabilization followed by the postoperative cohort. Although Group III in each cohort had superior ambulatory rates compared to Group I in each cohort, laminectomy plus vertebral stabilization followed by postoperative RT conferred higher rates of ambulatory recovery in all prognostic groups when compared with laminectomy and postoperative radiotherapy.

Recently, the same authors revisited the scoring system to assess if it was possible to select patients for upfront surgery based on data from 283 patients treated with radiotherapy alone in a prospective trial [74]. Multiple factors were collected, but only a few were considered prognostic factors of relevance based on a multivariable logistic regression model (i.e., tumor type, pre-radiotherapy ambulatory status, pre-radiotherapy sensory deficits, and pre-radiotherapy sphincter dysfunction) and after a backward stepwise variable selection technique. In addition, it was assessed for its internal validation and model performance measurements. The score point methodology for the selected relevant prognostic factors was calculated using the same approach as in 2008. In this recent study, only four prognostic factors of relevance were selected (Table 4). There were changes in the total score points (minimum 17 and maximum 37 points) and among the new 3 prognostic groups: 17–21 points for Group I, 22–31 points for Group II, and 32–37 points for Group III. The authors confirmed again the increment of postradiotherapy ambulatory rates (10% Group I, 65% Group II, and 97% Group III), as well as the two-year LC rates (100% Group III, 75% Group II, and 88% Group I). Additionally, the authors assessed the positive predictive values for ambulatory status in Group III and Group I which were 97% and 90% using the new score, compared to the 2008 criteria system for the same groups (98% Group III and 79% Group I). The authors present a new and improved scoring system for predicting non-ambulatory status, and also suggest that patients classified in the new Group III may not require surgery.

### 3.4. Clinical Criteria and Its Validation

The anatomical, biomechanical, and neurological complexity of the spine makes the management of spinal metastases particularly challenging in comparison with the approach of bone metastases located in other skeletal topographies [75].

There is a significant plethora of clinical criteria that should be taken into account and integrated in order to make a tailor-made and adapted decision.

The patient’s performance status, a measurement of its ability to perform certain activities of daily living without the help of others and a reflection of its general condition and fitness to tolerate the treatment, is pivotal and may be classified either by the Eastern Cooperative Oncology Group (ECOG) or by the Karnofsky scales.

The art of prognostication is difficult, and methodically establishing a 6-month expected survival is not simple. The Tomita and Tokuhashi modified scores, the SORG nomogram, and the New England Spinal Metastasis Score (NESMS) are useful and valuable tools to establish a prognosis, which is of paramount importance in order to decide what treatment option should be proposed.

Intrinsic characteristics of cancer, such as cancer burden—conceptually estimated by the size and location of the primary tumor, the number, size, and location of metastases, the clinical effects directly attributable to the primary tumor and metastases in the neighbor or distant tissues and organs, and by markers of cellular turnover such as lactate dehydrogenase (LDH)—stage, biological behavior and histology, are also determinants of different treatments responsiveness and decisively shape the patient’s prognosis. In some cases—such as Hodgkin and non-Hodgkin lymphomas, germ-cell neoplasms, myelomas, neuroblastomas, prostate, and breast cancers—a high chemo and/or radiosensitivity is verified [75]. For these cases, medical and/or radiation treatment should be proposed instead of surgery. On the other hand, other cases—such as non-small cell lung cancer, colon carcinoma, and carcinoma of unknown primary origin—display radio-resistance and show, in some series, short survival outcomes after spinal surgery, therefore benefiting less from an extensive intervention [75].

The clinical decision-making process is also constructed with the help of different frameworks such as the neurological, oncological, mechanical, and systemic (NOMS) framework, the already mentioned location, mechanical instability, neurology, oncology, and patient’s features (LMNOP) score, and also the Metastatic Spine Disease Multidisciplinary Working Group Algorithms (MSDA) [75].

The golden principle that must guide the consideration of every single feature, its integration in the clinical decision process, and, overall, the management of patients with spinal metastases is multidisciplinarity.

### 3.5. The Complexity of Management: Concrete Examples

Although systemic therapy does not usually play a part in the acute management of spinal-related adverse events, the medical oncologist plays an important role in defining patient prognosis and planning systemic treatment after patient stabilization. Local treatments directed to spinal metastases and spinal cord compression are rarely curative, and systemic treatment is almost always required in the setting of metastatic disease. The long-term prognosis of the patient is thus dependent on the available options for systemic treatment, their efficacy, and tolerability.

With recent advances in systemic treatment, many tumors that commonly metastasize to the spine have a good medium-term prognosis, even with metastatic disease, with overall survival measured in years. The role of the oncologist is thus essential when establishing a prognosis. For example, hormonal receptor-positive breast cancer has a median overall survival (OS) of 63.9 months with first-line hormone therapy and cyclin-dependent kinase inhibitors [76]. HER2-positive breast cancer has a median overall survival of approximately 5 years when treated with a combination of taxane-based chemotherapy and HER2 blockers [77]. Castration-sensitive prostate cancer boasts a median overall survival of > 53 months when treated with androgen-deprivation therapy and apalutamide [78]. Despite being notoriously resistant to classical chemotherapy, renal cell carcinoma is now treatable with double immunotherapy or with a combination of a tyrosine-kinase inhibitor (axitinib) and immunotherapy with pembrolizumab, with a median overall survival of 45.7 months [79]. Similarly, multiple myeloma has a good prognosis, even when metastatic bone disease is present [80]. One should notice that some of the first-line treatments for the cancers above can be initiated even in frail patients. Although immunotherapy trials frequently exclude patients with ECOG PS > 1, real-world evidence and clinical trials have shown these treatments to be safe in frail and elderly patients [14,81,82].

On a slightly different note, although an uncommon source of bone metastases, germline testicular tumors deserve to be mentioned, as they are extremely chemosensitive to platinum agents and are potentially curable even with distant metastases [83]. In this case, however, optimal treatment requires high-intensity chemotherapy and good performance status.

Prognostication in lung cancer is complex because it depends on histological subtypes (squamous cell carcinoma vs. adenocarcinoma vs. small-cell lung carcinoma), the presence or absence of targetable mutations (e.g., ALK fusion, EGFR exon 20-deletion), and markers of response to immunotherapy (such PD-L1 expression) [84].

The examples above help illustrate the complexity of prognostication and its possible impact on future decisions.

Table 4 summarizes examples of tumors with a good prognosis even in the setting of metastatic disease.

**Table 4 cancers-15-01796-t004:** Examples of tumors with a good prognosis, even in the setting of metastatic disease. Prognostication in lung cancer is complex because it depends on histological subtypes (squamous cell carcinoma vs. adenocarcinoma vs. small-cell lung carcinoma), the presence or absence of targetable mutations (e.g., ALK fusion, EGFR exon 20-deletion), and markers of response to immunotherapy (such PD-L1 expression) [84]. This table is not meant to be exhaustive. New developments continuously improve the prognosis for cancers of different types. The local availability of treatments must be taken into account in the prognostication of these tumors, especially in resource-poor countries.

Tumor Type	First-Line Treatment	Treatment Efficacy
Hormonal receptor-positive breast cancer	Combination of hormone-therapy and cyclin-dependent kinase 4/6 inhibitors (palbociclib, ribociclib, and abemaciclib)	Median overall survival (OS) is 63.9 months for ribociclib [76]
HER2-positive breast cancer	Combination of taxane-based chemotherapy and dual HER-2 blockade	Median OS is 56.5 months [77]
Castration-sensitive prostate cancer	Androgen deprivation therapy plus hormonal therapy	Median OS > 52 months (median overall survival not yet reached) [78]
Renal cell carcinoma	Combination of a tyrosine-kinase inhibitor (axitinib) and immunotherapy (pembrolizumab) or double immunotherapy (ipilimumab + nivolumab)	Median OS is 45.7 months [79]
Multiple myeloma	Bortezomib (proteasome inhibitor), lenalidomide, and dexamethasone (corticosteroid)	Median OS is 60 months [80]
Testicular germ cell tumors	Platinum-based chemotherapy (e.g., bleomycin + cisplatin + etoposide−good performance status only)	Good-risk disease: 90% cure rate; intermediate-risk disease: 80% cure rate; high-risk disease: 50% cure rate [83]

### 3.6. Objective Metrics to Appropriately Prognosticate, Judge, and Decide

The Tomita and Tokuhashi scores were originally considered the gold-standard tools for the prognostication of patients with spinal metastases, while the SORG nomogram represented a fresh and innovative strategy to tackle this issue, and, finally, NESMS took the spotlight as the score of choice. An individual, brief description is provided below.

The NOMS, LMNOP, and MSDA frameworks are widely used algorithms to support and structure the clinical decision-making process. A more detailed explanation may also be read below.

### 3.7. Prognostication

#### 3.7.1. Tomita Score

The Tomita score, introduced in 2001, is a prognostic and treatment-defining scoring system that categorizes cancers according to their grade, presence of visceral metastases, and number of bone metastases [85]. An individual score is attributed to each of these three factors, and a global prognostic score (between 2–10) results from the sum of each of these individual scores. The treatment goal and strategy for each patient is set according to this prognostic score: A prognostic score of 2–3 points suggests a wide or marginal excision for long-term local control; 4–5 points indicate marginal or intralesional excision for middle-term local control; 6–7 points justify palliative surgery for short-term palliation; and 8–10 points indicate nonoperative supportive care [85].

#### 3.7.2. Tokuhashi Score

The Tokuhashi score, firstly proposed in 1989 and later revised in 2005, takes a total of six variables into account, namely patient condition, number of extraspinal bone metastases, number of bone metastases in the vertebral bodies, resectability of metastases to major organs, site of primary cancer, and degree of subsequent paralysis [86,87]. In accordance with the 2005 revised version, each parameter ranges from 0 to 5 points, and the total score is 15 points [87]. Conservative treatment or palliative procedures are recommended for patients with a total score of 8 or less or for those with multiple vertebral metastases, while excisional procedures are recommended for patients with a total score of 12 or more or for those with a total score of 9 to 11 and with metastasis in a single vertebra [86,87].

Different series comparing the Tomita and the modified Tokuhashi scores accuracy in retrospectively determining prognosis for patients afflicted with spinal metastases have shown a higher prognostic power of both scores for patients with less than six months’ survival and a global better prognostic accuracy of the modified Tokuhashi score [88,89].

#### 3.7.3. Skeletal Oncology Research Group (SORG) Nomogram

A nomogram offers advantages over traditional methods of constructing a survival algorithm. These traditional methods are based on rounding the effect estimates (for example, hazard ratios) of prognostic factors in order to weigh up the present factors and relate them to survival estimates [90]. A nomogram describes more precisely the effect estimates on prognostic factors, offering a user-friendly tool that includes prognostic factors set to a common point scale [90]. Paulino Pereira et al. detected and highlighted risk factors that are independently associated with worse survival in patients with spine metastases—older age, poor performance status, specific primary cancers with poor prognosis, more than 1 spine metastases, presence of a lung, liver, and/or brain metastasis, previous systemic therapy, increased white blood-cell count, and decreased hemoglobin levels—and consequently developed and published a classic scoring system, a nomogram, and a boosting algorithm with proven accuracy in 2016 [90]. Further and posterior analyses revealed that the SORG nomogram and boosting machine-learning (ML) algorithm retrospectively predict 3- and 12-month survival in operated spinal metastatic patients with higher accuracy than the original and modified Tokuhashi scores [91]. The SORG ML algorithm has shown superior survival-predicting power in several recent series and studies and has been externally validated [91,92,93]. Additionally, the SORG nomogram seems to have the highest accuracy in retrospectively predicting 90-day and 1-year survival in surgically treated spinal metastatic patients among an array of different scoring systems, also including Tomita, modified Tokuhashi, modified Bauer, revised Katagiri, and van der Linden [94].

#### 3.7.4. New England Spinal Metastasis Score (NESMS)

The NESMS, originally published in 2015, was developed using 1-year mortality after metastatic spinal surgery as the sole outcome measure [95]. This score assigns points based on a patient’s modified Bauer score (the Bauer score attributes a classification on the basis of absence of visceral metastases, absence of lung cancer, presence of primary tumors of the breast or kidney or existence of lymphoma or myeloma, and presence of solitary skeletal metastases) (≤2 vs. ≥3), functional status (ambulatory vs. impaired or non-ambulatory for any reason, including neurologic, impairment or limitations due to pain), and preoperative serum albumin (<3.5 g/dL vs. ≥3.5 g/dL), with a maximal score of 3 [95].

NESMS has shown clinical accuracy for predicting short-term (30-day) major morbidity and mortality following metastatic spinal surgery [96].

The most differential and striking feature of NESMS is its prospective validation as a clinical prediction score for survival in patients with spinal metastases, in a study including 180 patients [97]. The study has shown that, compared to NESMS 3, those with a score of 2 had significantly greater mortality, as did those with a score of 1 [97]. A NESMS score of 0 was associated with a perfect prediction for 1-year mortality (100% of individuals with this score were deceased at 1-year) [97]. Similar results were found for mortality at 6-months and overall [97]. NESMS has shown utility in prognosticating survival for patients with spinal metastatic disease, irrespective of selected treatment strategy [97].

Due to its accuracy and prospective validation, we have chosen this score for inclusion in our treatment algorithm.

Despite their prognostic accuracy and clinical utility, one must always keep in mind that a common limitation in these scores is the fact that they are developed at a specific timepoint, framed by the treatment landscape of the moment of their development, not capturing the evolution of this treatment landscape and not taking into account new therapeutic options that are posteriorly developed—no scoring algorithm can be expected to remain relevant for long in the presence of systemic treatments developed after the algorithm’s publication.

### 3.8. Clinical Judgment and Decision

#### 3.8.1. Neurologic, Oncologic, Mechanical, and Systemic Status (NOMS) Framework

The NOMS framework development occurred in 2013. Its development process was an evidence-based medicine process, based on the available literature and expert consensus [98].

This clinical decision framework takes four characteristics into account, namely the presence of myelopathy and degree of epidural extension (classified accordingly with the SOSG scoring system definition), tumor radiosensitivity, degree of spinal instability (defined by the SINS score), and general systemic status and conditions to tolerate a surgical procedure, providing a decision framework that considers sentinel decision points in the treatment of spinal metastatic patients [98,99].

The neurological assessment (N) combines specific characteristics of the clinical presentation and imaging findings, such as the presence and severity both of clinical myelopathy and radiculopathy, as well as imaging evidence of the epidural spine cord compression score (ESCC) [31].

The oncologic (O) assessment evaluates the best method of achieving local tumor control, most often with radiation or systemic therapy [97].

The neurological and oncological considerations together determine the need for radiotherapy and/or surgery [31,98].

Mechanical instability (M) is assessed separately, and its definition is based largely on the presence of mechanical pain correlated with radiographic criteria embedded in the SINS [31]. As noted, the determination of instability defines the need for an interventional procedure. Radiotherapy, despite its ability to relieve pain in other contexts, is unable to relieve the pain from unstable fractures and is similarly incapable of restoring stability [31,98].

The assessment of systemic disease (S) and comorbidities is paramount in this approach. A patient who is too frail may be unable to tolerate an intervention or to derive a benefit from it [31,98].

#### 3.8.2. Location, Mechanical Instability, Neurology, Oncology, and Patient’s Features (LMNOP) System

The LMNOP system evaluates a number of variables: (L) Location of the disease in the spine and the number of spinal levels involved, (M) mechanical instability, (N) neurological signs and symptoms, (O) oncology-related factors (tumor type, radiosensitivity, etc.), and (P) patient-related factors such as patient fitness, prognosis, and response to prior therapy (P) [100].

Although not an actual treatment algorithm, LMNOP works as a general framework to help decide the most appropriate treatment and is a useful mnemonic for the key factors that must be assessed in order to define the most appropriate treatment for an individual patient with metastatic spine disease [100].

Location refers not only to the topographic levels of involvement (solitary vs. multilevel) but also the involvement of the anterior and/or posterior columns, and should be evaluated in detail [100]. The assessment of mechanical stability must be performed using SINS [100]. A complete neurological evaluation should also be performed in order to confirm or exclude the existence of symptomatic epidural cord compression [100]. It is also of paramount importance to estimate the degree of radiosensitiveness or radioresistance of the tumor [100]. Finally, defining patient fitness levels, properly drawing a prognostic horizon, and analyzing prior failed therapies also influence and shape the treatment decision [100].

In the LMNOP framework, patients with spinal cord compression may be offered open surgery depending on instability, prognosis, and patient fitness. Spinal instability without spinal cord compression may be treated with minimally invasive surgical procedures. For particularly radiosensitive tumors, external beam RT may be offered in monotherapy, even in the setting of spinal cord compression. In the case of radioresistant tumors, sterotactic radiosurgery is an attractive approach [92].

#### 3.8.3. Metastatic Spine Disease Multidisciplinary Working Group Algorithms (MSDA)

This algorithm has been developed using evidence from the literature and expert consensus. While highlighting the need for a multidisciplinary approach and stressing the role of interdisciplinary referrals, it provides clear and direct recommendations regarding the use of available treatment options [31].

Specific management algorithms are provided for five different clinical scenarios from the less severe asymptomatic spinal metastases or uncomplicated painful spinal metastases to the more severe stable pathologic vertebral compression fracture, unstable pathologic vertebral compression fracture, and metastatic epidural spinal cord compression [31]. Patients are initially stratified for each scenario by life expectancy, performance status, and number of visceral metastases [31]. Patients with a good prognosis (defined as having an overall survival of more than 6 months, a good performance status, and a low burden of visceral disease) are subsequently stratified according to the number of spinal metastases [31]. Particular therapeutic approaches are proposed for each sub-group of patients integrated in each scenario [31].

## 4. A New Integrative Flowchart Proposal

Metastatic spine disease is comprised of a wide variety of clinical manifestations with variations according to the primary neoplastic disease involved, the topography of spinal involvement, the presence or absence of spinal cord compression, and spinal stability (which can be defined by the SINS) [101].

As previously mentioned, there are some algorithms that guide decision-making in metastatic spine disease, namely the NOMS framework [102], LMNOP System [32] and MSDA [98] as there are certain scores that are useful for survival prediction, such as the Tomita score, the modified Tokuhashi score, the SORG nomogram, and NESMS [24].

However, it is crucial to note that the NOMS framework uses the SORG nomogram (which is not prospectively validated), the LMNOP system is a general framework and not an algorithm, and the MSDA is relatively complex and does not use a prospectively validated prognostication tool (such as NESMS). The override of prognostication tools by clinical judgement in cases of cancers for which the treatment is nowadays substantially better than it was at the time of the prognostication tool development is not foreseen in any of these clinical decision-making instruments. The quality and efficiency of the approach to metastatic spine disease patients tend to improve when a multidisciplinary vision and decision-making are adopted [103]. A flowchart regarding metastatic spine disease should ideally include medical management, such as hormonal therapies, chemotherapy, corticosteroid therapy, bisphosphonates, antiangiogenic agents, and denosumab; radiotherapeutic modalities, such as external beam conventional radiotherapy and stereotactic body radiation therapy (regarding this matter, the radiosensitivity of the tumor should be considered); open surgery (decompressive laminectomy, pedicle screw fixation, intersomatic device implantation, separation surgery); and minimally invasive approaches (vertebroplasty, kyphoplasty, ablative techniques, and percutaneous spine fixation).

We propose an innovative updated flowchart that, in our view, is a simpler and more user-friendly tool than the previous mentioned frameworks/systems to guide decision-making, with a dichotomized approach to two distinct clinical scenarios as opposed to multiple scenarios, which render previous scores harder to use.

This integrative approach conceptualization was primarily based on an extensive literature review and was enriched by the personal clinical experience of the experts at our center. Its development was gradual and resulted from the integration of the most updated data regarding the subject and different critical inputs and views of the experts at our center.

### 4.1. Goals of the Flowchart Development

Our objective is to define a set of guidelines to facilitate a multidisciplinary and systematic approach to patients with metastatic spine disease.

As previously mentioned, owing to the inherent heterogeneity of metastatic spine disease, the definition of an algorithm-based diagnostic and therapeutic approach is a complex task.

We aim to create a proposal for a multidisciplinary team-based approach to metastatic spine disease. In our center, we considered that the team should include oncologists, radiotherapists, neurosurgeons, orthopedic surgeons, hematologists, and neuroradiologists.

The design of the proposed flowcharts intends to avoid the time-crystallization phenomenon that is inherent to many of the previously described algorithms and flowcharts. Otherwise, our aim is to draw an evolution-sensitive and flexible tool that has the potential to keep accuracy and rigor when used in an unknown future timepoint with radically different options for systemic treatment (the definitions of the clinical scenarios and the emphasis put on the clinical judgement as the main criteria for the patients’ categorization within each of these scenarios are the epitome of this principle).

### 4.2. Clinical Scenarios and Proposed Flowcharts

The flowcharts in our proposal for decision-making regarding metastatic spine disease integrate multidisciplinary care, allowing for a quicker and more objective approach, which is paramount to achieving better clinical results.

Our proposal does not aim to lessen the importance of a personalized case-by-case analysis of each patient’s clinical status. Consequently, our goal is not to make the approach to metastatic spine disease rigid and inflexible.

We propose the division of patients with MSD into two clinical scenarios involving the Karnofsky Performance Status (KPS), expected 6-month survival according to the NESMS, and the status of neoplastic disease:Scenario 1 comprises patients with KPS ≤ 40% and/or expected 6-month survival < 50% and/or oncologic disease with multisystemic and progressive involvement and a lack of therapeutic options (as assessed by the medical oncologist).Scenario 2 includes patients with KPS > 40% and expected 6-month survival ≥ 50% and oncologic disease with stable/limited involvement or multisystemic involvement with available therapeutic options (as assessed by the medical oncologist).

The clinical scenarios were defined to facilitate the dichotomization of patients into two distinct groups that differ according to their performance status, predicted survival at 6-months based on the NESMS score, and status of neoplastic disease (encompassing the stage of systemic disease and the presence or absence of therapeutic options).

We chose to utilize the Karnofsky performance status as it is an ordinal scale which allows for a descriptive prediction of the clinical status of the patient and his capacity to perform ordinary tasks critical for daylife functionality. The 40% cutoff was defined as it signifies that the patient is bedridden for more than half of a day, translating into a clinical stage of oncologic disease in which the patient probably is not capable of tolerating more invasive or aggressive therapeutic modalities.

The NESMS scoring system was our choice in terms of prognostication tools owing to the fact the prospectively validated this score’s statistical significance as a tool to calculate survival at 6-months and 1-year, regardless of the therapeutic strategy chosen.

However, while prognostication using the NESMS score is advocated, the medical oncologist may override this score in the case of the existence of innovative treatments with good efficiency and tolerability which have appeared since the development and validation of the NESMS. The therapeutic approaches in oncologic patients are ever-evolving, and new efficient drugs are regularly being approved by regulating pharmaceutical associations.

Regarding the status of neoplastic disease, progressive disease with a lack of oncological therapeutic options should tend to deter the clinician from proposing the patient undergo more invasive therapeutic modalities which the patient possibly is not able to tolerate.

Our algorithm shares many similarities with the NOMS algorithm, upon which it was based. As in the NOMS algorithm, treatment is based on factors related to the tumor (in particular, to the prognosis of the systemic disease and radiosensitivity), to the patient (performance status and fitness for surgery), and to the degree of spinal cord compression and spinal instability. Similar to the NOMS algorithm and the LMNOP framework, it highlights the role of palliative surgery for pain control in spinal instability and refractory pain. Unlike the NOMS algorithm, we have chosen to use NESMS for prognostication instead of the SORG nomogram, as the former has been prospectively validated with excellent accuracy. Newly expected data regarding further validation of the SORG ML algorithm may also allow the possibility of using that specific tool for prognostication as part of these flowcharts.

We note that we propose the use of these algorithms as an aid to the clinical decision, not as a replacement of clinical judgement. The clinical judgement of the patient’s oncologist should always be taken into account, and prognosis-prediction scores must be used with caution as it is not possible to guarantee that a specific patient’s clinical scenario is completely foreseeable by the NESMS scoring system or other prognostication tools. Therefore, although our support for having the medical oncologist override the NESMS may be criticized as a way of introducing subjectivity into what is meant to be an objective decision, we claim that the use of expert opinion is advisable in this case until the NESMS has been validated in a more modern setting, taking into account new developments in systemic treatment.

Figure 2 and Figure 3 depict the proposed flowcharts in clinical scenarios 1 and 2, respectively.

## 5. Conclusions

Due to the improvement of medical and surgical management of oncologic patients, metastatic spinal disease is gradually becoming more prevalent in the current day [19]. This cohort of patients is inherently heterogeneous and complex. Therefore, a multidisciplinary personalized approach, including the expertise of each involved specialty (namely oncologists, radiotherapists, neurosurgeons, orthopedic surgeons, hematologists, and neuroradiologists), is crucial and achieves better results in terms of clinical outcomes [19].

In this multidisciplinary article, we present a review of the most recent data regarding the physiopathology of metastatic spinal disease, prognostic scores, treatment options, and we propose an updated algorithmic approach to the pathology according to the clinical scenario of each patient [27,32,96,97,98,102].

We propose a flowchart-based approach to patient management, which we believe will result in better results in an evidence-based management of metastatic spinal disease and spinal cord compression.

Nevertheless, we underline that the goal of this type of approach is to assist in clinical decisions and not to replace a case-by-case reflection concerning the specificities of each patient.

Metastatic spine disease management remains a palliative approach. Nonetheless, medical and surgical care are cardinal in ameliorating pain and improving quality of life in such patients [104].

## Figures and Tables

**Figure 1 cancers-15-01796-f001:**
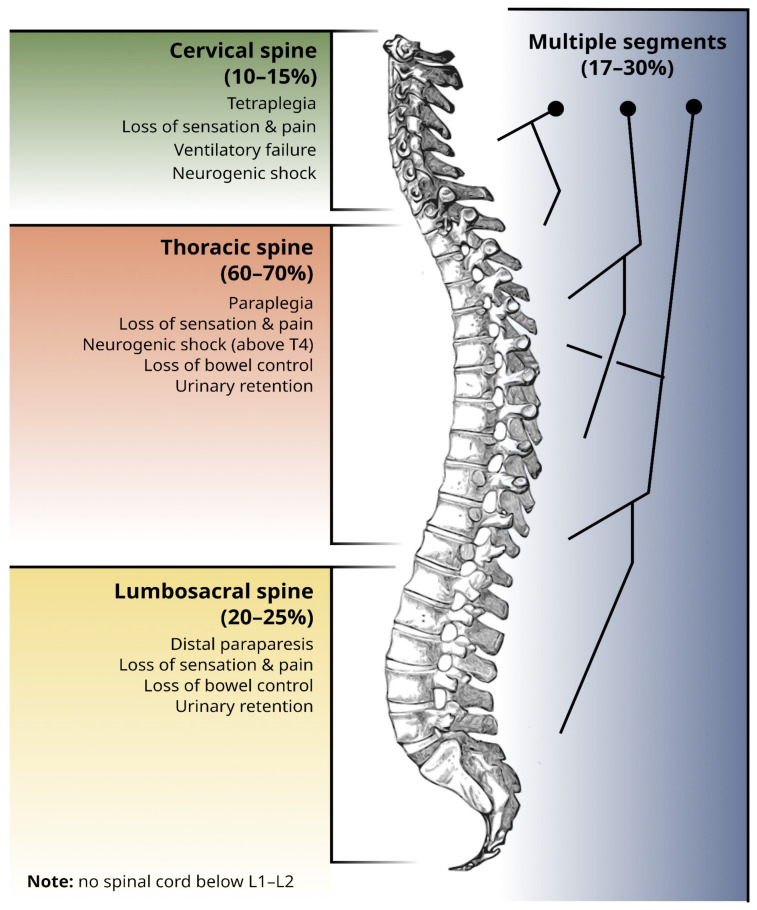
Proportion of metastases for each spine segment (cervical, thoracic, and lumbosacral), together with the most common clinical manifestations of neurological compromise at each level. Note that there is no spinal cord below the L1-L2 transition, which means that signs and symptoms from neurological compromise below this level are due to nerve root compression and not spinal cord compression. Percentages are according to [19]. Spine image reused from [20], in the public domain.

**Figure 2 cancers-15-01796-f002:**
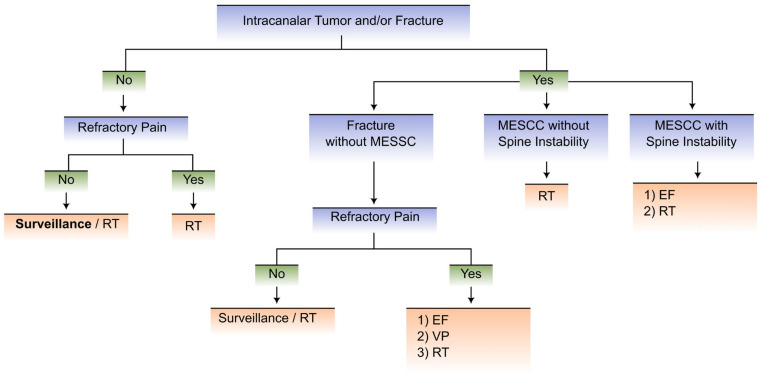
Flowchart for patients with clinical scenario 1 (KPS ≤ 40% and/or expected 6-month survival < 50% and/or oncologic disease with multisystemic and progressive involvement and a lack of therapeutic options). Labels: RT—radiotherapy; EF—external Fixation; MESCC—metastatic epidural spinal cord compression; VP—vertebroplasty.

**Figure 3 cancers-15-01796-f003:**
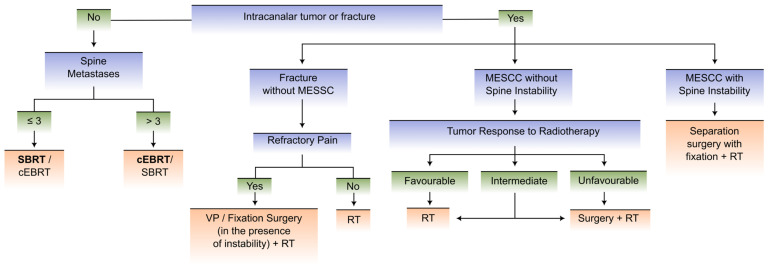
Flowchart for patients with clinical scenario 2 (KPS > 40%, expected 6-month survival ≥ 50%, and oncologic disease with stable/limited involvement or multisystemic involvement with available therapeutic options). Labels: cEBRT—conventional external beam radiation therapy; MESCC—metastatic epidural spinal cord compression; SBRT—stereotactic body radiation Therapy; RT—radiotherapy; VP—vertebroplasty.

**Table 1 cancers-15-01796-t001:** Multidisciplinary approach at three different moments related to metastatic spinal cord compression.

	Professional	Roles
**Before spinal cord compression**	Medical oncologist	Prognostication of the disease and definition of systemic treatment of both tumors and metastases (refer to the discussion of bone-targeted agents, as mentioned in Section 1.1); patient care coordination
	Radiation oncologist	Local treatment of symptomatic metastatic disease (and possibly the primary tumor, when indicated)
	Spinal ourgeon	Assessment of the risk of spinal cord compression; preemptive decompression or stabilization
	Radiologist	Assessment of the risk of spinal cord compression; diagnosis of the likely primary tumor; image-directed biopsy of the primary tumor or metastasis
	Pathologist	Histological diagnosis from tissue samples of the primary tumor or metastasis (essential for prognostication and systemic treatment)
	Palliative care specialist	Assessment and treatment of physical, emotional, and spiritual distress
**Acute spinal cord compression**	Assisting doctor in the emergency department	Often the first to encounter the patient, especially if the cancer is not yet known Diagnosis/suspicion of spinal cord compression and activation of the proper care pathways Organ support in the perioperative period (neurogenic shock and ventilatory failure)
	Medical oncologist	Prognostication of the disease (there is a very limited role for cancer-directed systemic therapy at this stage)
	Radiation oncologist	Metastases-directed treatment, with or without surgery
	Spinal surgeon	Spinal cord decompression, stabilization, and/or resection of the neoplastic lesion Can obtain specimen for histological diagnosis
	Radiologist	Differential diagnosis of the neurological syndrome
**After spinal cord compression (established neurological deficit and permanent disability)**	Medical oncologist	Treatment of the systemic disease to prevent further deterioration and increase survival and quality of life
	Radiation oncologist	Treatment directed to metastases or primary tumor (if indicated)
	Spinal surgeon	Decompression and stabilization to prevent deterioration, treat pain and allow for early mobilization Can obtain specimen for histological diagnosis if there is no diagnosis at this stage
	Rehabilitation specialist	Optimization of the remaining neurological function to increase patient autonomy and preserve quality of life Assistive devices to improve patient autonomy and safety
	Palliative care specialist	Assessment and treatment of physical, emotional, and spiritual distress related to the systemic disease and neurological complications

**Table 2 cancers-15-01796-t002:** Discrepancy of responsiveness to radiation and intrinsic radiosensitivity.

Responsive to Radiation	Resistant to Radiation
Lymphomas (high α/β) *	Malignant melanoma (low α/β)
Multiple myeloma (high α/β) *	Renal cell cancer (low α/β)
Small cell lung cancer (high α/β)	Non-small cell lung cancer (low α/β)
Germ cell tumors (high α/β) *	Gastrointestinal cancers (low α/β)
Prostate cancer (low α/β)	Sarcomas (low α/β **)
Breast cancer (low α/β)	

* Chemosensitive, ** Generalized for bone and soft-tissue sarcomas, with some exceptions (e.g., myxoid liposarcomas).

**Table 3 cancers-15-01796-t003:** Five prognostic factors and scoring system from Rades et al. 2008 [72].

Variable	Post-RT Ambulatory Rate, Percent	Score
Type of primary tumor		
Breast cancer	81	8
Prostate cancer	68	7
Myeloma/lymphoma	89	9
Non-small cell lung cancer	54	5
Small cell lung cancer	64	6
Cancer of unknown primary	45	5
Renal cell carcinoma	62	6
Colorectal cancer	64	6
Other tumors	59	6
Interval from tumor diagnosis to MSCC		
≤15 months	58	6
>15 months	78	8
Motor function before RT		
Ambulatory without aid	98	10
Ambulatory with aid	89	9
Not ambulatory	28	3
Paraplegic	7	1
Time of developing motor deficits before RT		
1–7 days	37	4
8–14 days	69	7
>14 days	88	9

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
