# Peer review of "Multidisciplinary Approach to Spinal Metastases and Metastatic Spinal Cord Compression—A New Integrative Flowchart for Patient Management"

_cancers, 2023, doi:10.3390/cancers15061796_

Round 1
Reviewer 1 Report
The manuscript on this interesting topic should have a major revision especially shortening and rounding. There are a lot of redundancies and repetitions, which makes it hard to read. When reading the manuscript one has the feeling of a symposium with consecutive presentations - each with a separate introduction -slides and conclusions.
Some citations are misleading, eg the radiation dose in the PRE-MODE Trial was not 5 Gy in 5 Fractions but 5 x 5 Gy (=25 Gy) as well as the historical control group 4 x 5gy(=20Gy)
I miss a chapter on the systemic treatment with rank ligand or bisphosphonates - a treatment frequently used in spine metastases.
Author Response
“The manuscript on this interesting topic should have a major revision especially shortening and rounding. There are a lot of redundancies and repetitions, which makes it hard to read. When reading the manuscript one has the feeling of a symposium with consecutive presentations - each with a separate introduction -slides and conclusions”.
Answer: We thank this reviewer for the time set aside to improve our manuscript and for the thoughtful review.
We have improved upon its feedback, which we feel has augmented the quality of the article. Globally, we have tried to rework the language and simplify the text.
Besides this, we agree that some redundancies were present, and that the previous repetitive pattern, instead of clarifying our main points, muddled the presentation of the content and negatively affected the intelligibility of the text. We have rewritten specific parts of the article, eliminating redundancies in order to improve the flow of ideas. We have removed some introductory sections which merely re-iterate material that is expanded later. All the modifications and eliminations are shown in track-changes, except for one paragraph (originally located in page 29, lines 596-601) that was eliminated. This paragraph’s elimination had to be accepted in the document to allow references to be assembled and integrated in the reference list by the bibliography software package.
We tried to harmoniously integrate the new contents (whose inclusion was suggested by the Reviewers) in the organic structure of the manuscript.
“Some citations are misleading, eg the radiation dose in the PRE-MODE Trial was not 5 Gy in 5 Fractions but 5 x 5 Gy (=25 Gy) as well as the historical control group 4 x 5gy(=20Gy)”
Answer: We thank this specification to Reviewer 1 and have adopted the clearer language that is suggested, in order to make it clearer that the total dose is 25Gy and 20Gy respectively. The modified version of this specific citation may be read on page 11, lines 545-561.
“I miss a chapter on the systemic treatment with rank ligand or bisphosphonates - a treatment frequently used in spine metastases”
Answer: Once again, we thank the reviewer for this helpful suggestion. We agree and we have accordingly introduced new content regarding the systemic treatment with rank ligand and bisphosphonates, as it may be read on page 3, lines 120-137.

Reviewer 2 Report
Comprehensive article, clearly written and easily understood; figures and flowcharts also clear. I believe it can be published.
Author Response
“Comprehensive article, clearly written and easily understood; figures and flowcharts also clear. I believe it can be published.”
Answer: We thank this reviewer for the kind assessment and for the time spent reading and reviewing our manuscript.
Reviewer 3 Report
Dear authors, thank you for this interesting article summarising different approaches and providing a new integrative approach. The latter topic is where I struggle with: how did the authors come up with this integrative approach? Based on literature? On expert opinion? For example, the NOMS framework is developed as the authors state through a “evidence-based medicine process, supported by the available literature and expert opinion consensus”. It is unclear through which process the authors come up with the new decision framework and this needs further explaining.
Other minor comments:
- SORG developed a machine learning prediction model that has been validated multiple times and demonstrates better performance than the nomogram. The nomogram is therefore outdated.
- Can any comments be made where future studies should focus on? What caveats remain in the literature that we need to investigate to improve medical decision making?
- Why is there a need to develop a new decision framework and not use current frameworks such as the NOMS? (I reason that they are outdated but please mention somewhere more clearly the rationale for developing a new approach)
- If word count allows it, can a short section be devoted to all the complications related to the different therapies? One must way the benefits with the potential complications.
Author Response
“Dear authors, thank you for this interesting article summarizing different approaches and providing a new integrative approach. The latter topic is where I struggle with: how did the authors come up with this integrative approach? Based on literature? On expert opinion? For example, the NOMS framework is developed as the authors state through a “evidence-based medicine process, supported by the available literature and expert opinion consensus”. It is unclear through which process the authors come up with the new decision framework and this needs further explaining”.
Answer: We thank the reviewer for this critical, but very correct assessment.
The section 4 was reformulated considering Reviewer 3’s comments.
The basis for the development of this integrative approach is now explicitly displayed on page 20, lines 1180-1188.
The purpose of the development of this new integrative approach and respective flowcharts is further explored on page 20, lines 1200-1206.
The methodological process for the development of this new decision framework is thoroughly explained on page 21, lines 1226-1269. The particular features and the conceptual strengths of our approach are also highlighted in that section.
“SORG developed a machine learning prediction model that has been validated multiple times and demonstrates better performance than the nomogram. The nomogram is therefore outdated”.
Answer: We thank the reviewer for this helpful suggestion. Accordingly, section 3.7.3 was reformulated and additional information regarding this machine learning prediction model and its value may now be read on page 17, lines 842-847.
“Why is there a need to develop a new decision framework and not use current frameworks such as the NOMS? (I reason that they are outdated but please mention somewhere more clearly the rationale for developing a new approach).”
Answer: We, once again, thank the reviewer for this precious assessment. The conceptual fragilities, in our view, of the current frameworks are now highlighted on page 19-20, lines 1072-1169. The advantages of our decision framework, in comparison with the current frameworks, are also displayed on page 20, lines 1180-1183 and lines 1200-1206, and page 21, lines 1249-1269.
“If word count allows it, can a short section be devoted to all the complications related to the different therapies? One must way the benefits with the potential complications.”
Answer: We thank the reviewer for this suggestion. We have included a new section dedicated to complications, as it may be read on page 9, lines 408-430. We highlight that this new section regarding treatment complications is highlighted in green to mark its introduction, as the changes had to be accepted in this document’s track-changes system in order to allow references to be assembled and integrated in the reference list by the bibliography software package.
Round 2
Reviewer 1 Report
Your work has substantially improved the manuscript, however it is still quite a long manuscript. You should check shortening eg the paragraphs lines 278-298 and 302-335 and 505-547, 764-798. There are repeated sections on "clinical criteria" confusing the reader.
Please check line 124 "R" missing in RANK
Line 131: In my experience its not hypercalcaemia but hypocalcaemia
Author Response
Reviewer 1
“Your work has substantially improved the manuscript, however it is still quite a long manuscript. You should check shortening eg the paragraphs lines 278-298 and 302-335 and 505-547, 764-798. There are repeated sections on "clinical criteria" confusing the reader”.
Answer: We thank reviewer 1 for the kind assessment and for the time spent reading and reviewing our manuscript.
The paragraph originally located between lines 278-298 was significantly shortened, as it may be seen in this new version of the manuscript. Besides this, the same was done for the paragraph originally located between lines 302 and 335 (now located between lines 304 and 318), while the table that was located between lines 505-547 was eliminated and the paragraph originally located between lines 764-798 was also significantly shortened.
The “3.4. Clinical criteria and its validation” was also very significantly shortened, and modified in order to be less descriptive and less redundant giving the subsequent detailed presentation of different features of each specific score and framework.
“Please check line 124 "R" missing in RANK”
Answer: We thank reviewer 1 for spotting this typo. The mistake was corrected as it may be read on line 123.
“Line 131: In my experience its not hypercalcaemia but hypocalcaemia”
Answer: We thank reviewer 1 for pointing out this error. This was naturally modified to hypocalcemia as it may be read on line 130.

Reviewer 3 Report
Dear authors, thank you for your revisions. If I understand correctly, the new proposed framework is based on literature and expert knowledge. I believe that new frameworks such as these should be established by panel discussion with multiple experts across different disciplines (as for example has been done with the NOMS framework).
Author Response
Reviewer 3
“Dear authors, thank you for your revisions. If I understand correctly, the new proposed framework is based on literature and expert knowledge. I believe that new frameworks such as these should be established by panel discussion with multiple experts across different disciplines (as for example has been done with the NOMS framework)”.
Answer: We thank reviewer 3 for his precious assessment.
We understand and share reviewer 3 conceptual concerns regarding the development and validation of a framework, and we salute it for its comment.
Our framework was originally drawn and progressively polished by a multidisciplinary panel of medical oncologists, neurosurgeons, and radiation oncologists. It resulted from the close interaction between these specialties in our center and it arose from the detailed discussion of several different patients affected by such an heterogenous and complex pathological context. Besides this, our framework was built using scores, scales and different types of metrics that are validated and is supported by the literature, as we explain throughout the text of this present paper.
We have been applying our framework, in parallel with the application of the traditional and validated frameworks and scores, as a potential management defining tool for patients affected by metastatic spine disease and metastatic spinal cord compression at our center. We are working on the estimation of degree concordance between our framework and the most currently used ones nowadays, and we are compiling data regarding survival metrics and quality of life variables of all the patients that have been managed by our team after the development of our framework and we are planning to publish these data relatively soon.
Apart from this, the publication of this paper serves the purpose of dissemination of our framework, a factor of paramount importance precisely for a broader discussion involving multiple experts across different disciplines, allowing its appreciation and potential establishment by a larger panel of experts.

Round 3
Reviewer 3 Report
Thank you for your explanation.